# In-Host Flat-like Quasispecies: Characterization Methods and Clinical Implications

**DOI:** 10.3390/microorganisms12051011

**Published:** 2024-05-17

**Authors:** Josep Gregori, Sergi Colomer-Castell, Marta Ibañez-Lligoña, Damir Garcia-Cehic, Carolina Campos, Maria Buti, Mar Riveiro-Barciela, Cristina Andrés, Maria Piñana, Alejandra González-Sánchez, Francisco Rodriguez-Frias, Maria Francesca Cortese, David Tabernero, Ariadna Rando-Segura, Tomás Pumarola, Juan Ignacio Esteban, Andrés Antón, Josep Quer

**Affiliations:** 1Liver Diseases-Viral Hepatitis, Liver Unit, Vall d’Hebron Institut of Research (VHIR), Vall d’Hebron Barcelona Hospital Campus, Passeig Vall d’Hebron 119-129, 08035 Barcelona, Spain; sergi.colomer@vhir.org (S.C.-C.); marta.ibanez@vhir.org (M.I.-L.); facedgc@gmail.com (D.G.-C.); carolina.campos@vhir.org (C.C.); mariaasuncion.buti@vallhebron.cat (M.B.); mar.riveiro@vallhebron.cat (M.R.-B.); david.tabernero@vhir.org (D.T.); juanignacio.esteban@vallhebron.cat (J.I.E.); 2Centro de Investigación Biomédica en Red de Enfermedades Hepáticas y Digestivas (CIBEREHD), Instituto de Salud Carlos III, Av. Monforte de Lemos, 3-5, 28029 Madrid, Spain; frarodri@gmail.com (F.R.-F.); maria.cortese@vhir.org (M.F.C.); ariadna.rando@vallhebron.cat (A.R.-S.); 3Biochemistry and Molecular Biology Department, Universitat Autònoma de Barcelona (UAB), Campus de la UAB, Plaça Cívica, 08193 Bellaterra, Spain; tomas.pumarola@vallhebron.cat; 4Medicine Department, Universitat Autònoma de Barcelona (UAB), Campus de la UAB, Plaça Cívica, 08193 Bellaterra, Spain; 5Centro de Investigación Biomédica en Red de Enfermedades Infecciosas (CIBERINFEC), Instituto de Salud Carlos III, Av. Monforte de Lemos, 3-5, 28029 Madrid, Spain; cristina.andresverges@vallhebron.cat (C.A.); maria.pinana@vallhebron.cat (M.P.); alejandra.gonzalez@vallhebron.cat (A.G.-S.); andres.anton@vallhebron.cat (A.A.); 6Microbiology Department, Vall d’Hebron Institut de Recerca (VHIR), Vall d’Hebron Barcelona Hospital Campus, Passeig Vall d’Hebron 119-129, 08035 Barcelona, Spain; 7Biochemistry Department, Vall d’Hebron Institut de Recerca (VHIR), Vall d’Hebron Barcelona Hospital Campus, Passeig Vall d’Hebron 119-129, 08035 Barcelona, Spain; 8Basic Science Department, International University of Catalonia, Sant Cugat del Vallès, 08195 Barcelona, Spain

**Keywords:** viral treatment failures, resistance-associated mutations (RAS), mutagens, enhanced fitness, flat-like quasispecies, near-flat, regular quasispecies

## Abstract

The repeated failure to treat patients chronically infected with hepatitis E (HEV) and C (HCV) viruses, despite the absence of resistance-associated substitutions (RAS), particularly in response to prolonged treatments with the mutagenic agents of HEV, suggests that quasispecies structure may play a crucial role beyond single point mutations. Quasispecies structured in a flat-like manner (referred to as flat-like) are considered to possess high average fitness, occupy a significant fraction of the functional genetic space of the virus, and exhibit a high capacity to evade specific or mutagenic treatments. In this paper, we studied HEV and HCV samples using high-depth next-generation sequencing (NGS), with indices scoring the different properties describing flat-like quasispecies. The significance of these indices was demonstrated by comparing the values obtained from these samples with those from acute infections caused by respiratory viruses (betacoronaviruses, enterovirus, respiratory syncytial viruses, and metapneumovirus). Our results revealed that flat-like quasispecies in HEV and HCV chronic infections without RAS are characterized by numerous low-frequency haplotypes with no dominant one. Surprisingly, these low-frequency haplotypes (at the nucleotide level) exhibited a high level of synonymity, resulting in much lower diversity at the phenotypic level. Currently, clinical approaches for managing flat-like quasispecies are lacking. Here, we propose methods to identifying flat-like quasispecies, which represents an essential initial step towards exploring alternative treatment protocols for viruses resistant to conventional therapies.

## 1. Introduction

The persistence or resolution of a viral infection, either by natural self-mechanisms or following antiviral treatment, is dependent on viral and host determinants [1,2,3]. Antiviral resistance is influenced by several factors, including the selection of resistance-associated substitutions (RAS) [1,4], mutations impacting the polymerase fidelity in response to treatment with mutagens [5], viral replicative fitness [6,7], the occurrence of some repeated amino acid substitutions other than RAS [1], and immunosuppression in patients [8]. A common thread is the dynamic adaptation of viral quasispecies to a shifting environment associated with the modification of the viral mutant spectrum. One such modification may be akin to the theoretical concept of flat quasispecies proposed decades ago with digital organisms [9]. One of the major consequences of viruses would be to confer a selective advantage in sequence space exploration in the absence of fitness peaks and valleys. A flat quasispecies may materialize as myriads of replicators forming a cloud of mutants occupying a substantial portion of the genetic space that is functional to the virus, all of which facilitate virus survival and evolution [9,10,11]. A flat quasispecies is expected to select the escape mutants that are present or accrue within a population as a consequence of the ease of genomic sequence movements in sequence space. This may have relevant clinical implications.

For hepatitis viruses, individuals with higher baseline viral loads or more advanced liver disease may be at a higher risk of treatment failure than other chronically infected patients. Long-lasting infections are associated with higher quasispecies fitness. Therefore, patients who have been chronically infected patients for a long time are at higher risk of treatment failure [12,13].

A recent study involving a patient with HEV treated repeatedly with increasing doses of ribavirin [14] shows an evolution towards flat-like quasispecies with enhanced fitness in the final resistant viral population. We hypothesize that the quasispecies of chronic patients with long-lasting infections and those resulting from treatment failures without known resistance mutations may generally exhibit attributes of flat quasispecies associated with high viral fitness, which we term flat-like quasispecies. Besides presenting empirical evidence of in-host quasispecies with this structure, we introduce specific methods for their characterization.

According to the respective definitions, quasispecies flatness and population evenness are closely related concepts, although they are not equivalent. Flatness refers specifically to the absence of significantly dominant genomes (displaying significantly higher fitness than the surrounding mutant cloud), whereas evenness emphasizes the uniformity in the frequency of different variants. High-level evenness means that all the components are roughly equally abundant, while low-level evenness indicates a more uneven or skewed distribution. A quasispecies with a high level of evenness may be qualified as flat-like only if (a) it has a high number of haplotypes, (b) there is no significantly dominant haplotype or a small subset of dominant haplotypes over the others, and (c) it occupies a significant fraction of the functional genetic space of the virus.

In this study, we confront three different models of quasispecies. A model of flat-like quasispecies was observed in the failed treatments of two chronic HEV immunocompromised patients, repeatedly treated with different regimens of ribavirin [14,15], consisting of numerous low-frequency haplotypes without a dominant one. The model of near-flat quasispecies was developed using data from patients with HCV failing direct-acting antiviral (DAA)-based treatments, characterized by having a master haplotype with moderate to low frequency. On the other site, regularly structured quasispecies (regular quasispecies) samples were found in patients with acute infectious respiratory viruses (betacoronaviruses, enterovirus, respiratory syncytial viruses, and metapneumovirus), consisting of populations with a prominent master haplotype [16].

## 2. Materials and Methods

The samples were collected from eleven patients: one chronically infected with subtype 1b of HCV [1], two patients chronically infected with HEV [15], and eight with various respiratory viruses. For the two patients with HEV, sequential samples from S10 to S13 were taken from one patient, and the samples from S14 to S17 were taken from another. No coinfections were observed in any of the patients included in this study. Additional samples from the patients with chronic HCV infection were analyzed in the Appendix A.

### 2.1. Regular Quasispecies Samples

Samples of the following viruses were collected to ensure easy availability and sufficient viral load for amplification with minimum bias, representing random RNA quasispecies. The amplicons correspond to the respective envelope or surface genes and are of similar length, ranging from a minimum of 317 base pairs (bp) to a maximum of 433 bp, with a median of 362 bp. They were processed simultaneously and sequenced in the same MiSeq Illumina run (Table 1).

### 2.2. HEV Flat-Like Quasispecies Samples

In the study, we included one amplicon of ORF2 (core gene) from sequential samples of two HEV chronic immunocompromised patients who were repeatedly treated with a mutagen (ribavirin) with different regimens. These two patients were unsuccessfully treated and exhibited flat quasispecies characteristics [14,15]. The primers and positions are provided in Appendix A. The follow-up of the two patients with HEV show very dissimilar results along their respective quasispecies evolution. These samples were processed in parallel for each run, showed high-quality sequencing scores, and included samples of a very different nature, thus excluding artifacts that could bias the results towards flat characteristics.

### 2.3. HCV Near-Flat Quasispecies Samples

The samples collected from six patients with HCV who failed treatments with DAAs showed similar near-flat quasispecies structures when studying the NS5B amplicon spanning positions from 7971 to 8366 (Appendix A). The results are presented in the Appendix A.

### 2.4. Sample Preparation

Viral RNAs were extracted from 140 µL of sample (serum or nasopharyngeal swab) via manual extraction using the QIAmp Viral RNA Mini Kit (Qiagen, Hilden, Germany) following the manufacturer’s protocol, but no RNA carrier was added during sample lysing. RNA was added to 30 µL of Elution Buffer. For each region, reverse transcription at 50 °C for 30 min and the first cDNA amplification (PCR) of 35 cycles at 55 °C were performed using the OneStep RT-PCR Transcriptor Kit (Roche Applied Science, Basel, Switzerland).

The PCR products were analyzed using electrophoresis on 1.5% agarose gel (Agarose MP, Roche Indianapolis, IN, USA), and the expected size-specific DNA bands were extracted from the gel and purified using the QIAquick Gel Extraction Kit (Qiagen, Valencia, CA, USA). The amplified and purified DNA was quantified using fluorescence using the Quant-iT Qubit dsDNA BR Assay Kit (ThermoFisher Scientific, Waltham, MA, USA). All the amplicons were normalized to the same concentration and purified using KAPA Pure Beads magnetic beads (KAPA Biosystems, Roche, Pleasanton, CA, USA). A second round of quantification via fluorescence was performed using the Quant-iT Qubit dsDNA HS Assay Kit (ThermoFisher Scientific, Waltham, MA, USA), followed by a second round of normalization of all the DNA pools to 1.5 ng/µL.

For library preparation for MiSeq, we used the KAPA HyperPrep Kit (Roche Applied Science, Pleasanton, CA, USA) standardized protocol with the SeqCap Adapters A/B (Nimblegen, Roche Pleasanton, CA, USA) to mark each sample with a specific index. A second clean-up with KAPA Pure Beads (KAPA Biosystems, Roche, Pleasanton, CA, USA) was performed to remove small DNA fragments that could contaminate the samples. Quality testing was performed with the Agilent DNA 1000 kit and bioanalyzer or 4200 TapeStation System from Agilent (Santa Clara, CA, USA).

The samples were normalized for the last time at 4nM before mixing 10 µL of each in a library tube. The final library was quantified via qPCR using the KAPA Library Quantification Kit (KapaBiosystems, Roche, Pleasanton, CA, USA) with a LightCycler480 (Roche) to obtain exactly the concentration of indexed DNA. The last rounds of dilution and mixing with an internal DNA Control (PhiX V3, Illumina, San Diego, CA, USA) were performed before loading the library to the MiSeq Reagent Kit 600V3 cartridge (Illumina, San Diego, CA, USA) and sequenced using the MiSeq platform (Illumina, San Diego, CA, USA). The flow cell was loaded with 600 μL of the library (containing PhiX) at 20 pM of denatured DNA final concentration, yielding 7.2 × 10^9^ single-strand molecules per forward and also per reverse.

### 2.5. Deep Sequencing

To acquire a comprehensive picture of quasispecies composition, our previous results emphasized the necessity of conducting deep-sequencing studies with very high coverage (over 1 × 10^5^ reads per amplicon) and avoiding unnecessary abundance-based filters [17]. This is particularly crucial with mutagenic treatments, where their effect in the short term is primarily observed at the lowest haplotype frequency [15,17]. Nevertheless, in cases of extreme complexity, the simplification introduced by filtering all the haplotypes below a modest number of reads, i.e., from 2 to 5, could be acceptable, provided that the resulting loss of information is limited.

### 2.6. Data Treatment of Fastq Files to Amplicon Haplotypes

The fastq files from the sequencer were treated to preserve full read integrity, completely covering the amplicon, to obtain amplicon-haplotypes and corresponding frequencies. Briefly, as previously described [14], full amplicon reads were obtained from the 2 × 300 bp paired-end reads with the help of FLASH [18], requiring a minimum overlap of 20 bp and a maximum of 10% mismatches; the reads accumulating more than 5% bp with Phred scores below Q30 were removed. The clean amplicon was finally obtained by trimming primers. Reads were collapsed into haplotypes and counts. These haplotypes and frequencies are the basis of subsequent computations. In previous studies, we intersected the haplotypes observed on both forward and reverse strands to obtain high-quality amplicon haplotypes. However, with flat-like quasispecies, this approach is unfeasible due to the low read fractions shared between the strands, indicating that we are in front of flat-like quasispecies. Hence, the results were computed for each strand independently, without previous abundance filtering or strand intersection.

### 2.7. Quasispecies Fitness Partition

Taking haplotype fitness into account, the reads sampled from quasispecies may be summarized in the form of four fractions [18]: The fraction corresponding to the dominant haplotype (master); the fraction of haplotypes with intermediate frequencies competing with the master, which we take as those over 1% in frequency (emerging); a fraction of low fitness, as the aggregation of reads corresponding to haplotypes with frequencies in the range 0.1–1% (rare); finally, the fraction with lowest fitness (very rare).

### 2.8. Hill Numbers

Quasispecies diversity is computed here in the form of Hill numbers [19]. By varying the order q from 0 to infinity, we obtain a characteristic profile [15], where the value at *q* = 0 corresponds to the number of haplotypes, *q* = 1 is the exponential of the Shannon entropy, *q* = 2 is the inverse of the Simpson’s index, and *q* = ∞ is the inverse of the master haplotype frequency.

(1)
$$D(P,q)={\left(\sum_{i=1}^{H}p_{i}^{q}\right)}^{1/\left(1-q\right)}$$


### 2.9. Indices of Evenness and Flatness

The Hill numbers [15,19] profile of a fully even quasispecies is a horizontal line at the level of the number of haplotypes in the quasispecies. All the haplotypes show the exact same frequency. In dividing a Hill number of order *q* by the Hill number of order 0, we can obtain a measure of evenness of the form *E = H/H*_max, where *H* represents the diversity index and *H*_max denotes the maximum possible value of *H*; this may be generalized to obtain a Hill evenness index of order *q* (Equation (2)) and, consequently, a Hill evenness profile, where all the curves start from the ordinate of 1, or 0 if represented in the log scale.

(2)
E(P,q)=D(P,q)/D(P,0)


As the highest drop between consecutive integer values of *q* in the profile of regular quasispecies is from *q* = 0 to *q* = 1, *E*(*P*,1) provides a useful measure of quasispecies evenness. Some interesting complementary values are *E*(*P*,2) and log10(*D*(*P*,*∞*))/log10(*D*(*P*,0)) (Equation (3)).

(3)
$$\eta (P,\infty )={log}_{10}\left(D(P,\infty )\right)/{log}_{10}(D(P,0))$$


These three indices are bound to values between 0 and 1. The higher the value is, the more even the distribution of haplotypes in the quasispecies is. Note that *D*(*P*,0), the number of haplotypes, is highly influenced by the sampling effort and is affected by a high sampling variability; this variability is transmitted to all the *E(P*,*q*) values.

Different alternatives [20,21] have been proposed for Equation (2) following this scheme. Among them, it is worth mentioning relative logarithmic evenness (RLE), which is expressed in Equation (2) after taking in to account the logarithms of the numerator and denominator. The RLE profiles have values in the range 0–1. Another alternative single index of evenness proposed is the Williams’ E9 index [22], with optimal properties based on the Euclidean distances between the given haplotype distribution and the two most extreme cases with the same number of haplotypes: perfect dominance and evenness. Nevertheless, this index seems to saturate easily with very deep quasispecies NGS data. One plausible cause is the curse of dimensionality [23], implied by the very high number of haplotypes observed in near-flat and flat-like quasispecies.

A complementary measure is the number of haplotypes, *D*(*P*,0), which represents the height of the diversity profile. The higher the profile is, the flatter the quasispecies is. A quasispecies with a modest number of haplotypes cannot be considered flat by any means, even if all are equally fit and even.

A different kind of index, measuring significant dominance in flat-like quasispecies, is obtained when considering the fraction of reads corresponding to a given number, *N*, of haplotypes at the highest frequencies or the ratio between this fraction and the frequency of the master haplotype (Equation (4)).

(4)
$$F_{topN}=\left(\sum_{i=1}^{N}p_{i}\right)/p_{1}$$


In this equation, *p_i_* represents haplotype frequencies arranged in decreasing order of value, and *N* denotes the number of top haplotypes to be considered. This index is bound between 1 and *N*. The higher its value is, the less prevalent the master haplotype is. We suggest 25 as a useful value for *N*.

### 2.10. Substitution Load

The portion of available genetic space explored and occupied is another characteristic of flat-like quasispecies [11]. Since the genetic space available to a virus is a priori unknown, the plasticity of a quasispecies in genetic terms may be indirectly quantified by the average substitution load by read. This value is obtained by calculating the weighted sum of substitutions across all the haplotypes relative to the master haplotype. The weights are determined by the number of reads supporting each haplotype. According to this definition, the master haplotype has 0 substitutions in this weighted sum.

### 2.11. Synonymity

The exploration of the genetic space by the selected HEV quasispecies with flat-like characteristics resulted in the selection of a significant number of haplotypes with synonymous mutations expressing the same phenotypes [11,14]; this corresponds to the selection of different haplotypes with the same level of fitness, which may contribute to a higher average fitness in the quasispecies, and possibly to a higher robustness against environment changes. The synonymity in a quasispecies may be easily evaluated by the ratio of the frequencies of the master phenotype to the master haplotype or, more generally, by the ratio of the aggregated frequencies of the top *N* phenotypes to the top *N* haplotypes, where *N* may be taken, i.e., as 5, 25, or 250 [11,14].

### 2.12. Rarefaction

The normalization of quasispecies diversity values to the sample size within this study was conducted through repeated subsampling without replacement to the reference size. At each resampling cycle, the diversity values were computed. After the *B* resampling cycles (here, 500), a median was computed and taken as the normalized value. The reference size represents the minimum acceptable size. Any sample with a size below the reference was excluded. In this study, the reference size, considering the observed coverages, was set at 194,000 reads.

## 3. Results

The samples were labeled with unique symbols, ranging from S01 to S19 (Table 1). In the figures and tables, these IDs are prefixed with the virus type for clarity. We studied the segments of genes encoding for external proteins/glycoproteins subjected to similar immune selection pressures of respiratory viruses such as betacoronaviruses (HKU1, NL63, OC43, and SARS-CoV-2), enterovirus A71, metapneumovirus B, and respiratory syncytial virus (RSV) A and B, as well as hepatitis C and E viruses. S18 and S19 are replicates of S16 and S17 with less coverage, and these were excluded from certain calculations. The primers for each virus and amplicon are reported in Appendix A.

### 3.1. Sequencing Coverage

Deep sequencing was performed using a MiSeq Illumina instrument, obtaining deep coverages ranging from 60,000 to more than 200,000 reads per strand, amplicon, and sample.

Despite that the values observed in both the strands for all the indices are very close, they are based on different sets of haplotypes for flat-like quasispecies, as indicates the poor intersection of haplotypes in the forward and reverse strands (Appendix A). Both are representative of the complete quasispecies and exhibit the same structure but correspond to different subsamples of the same highly complex population. The dissimilar results are the consequence of dissimilar coverages in both strands, likely caused by dissimilar efficiencies of the forward and reverse specific primers. On the other hand, the poor intersection implies that the coverage obtained for these samples, ~400,000–500,000 reads/amplicon, is insufficient to account for all the variants present in flat-like quasispecies.

The basic calculations clarify this observation: a MiSeq flow cell can generate about 20 million reads after being fed with 600 μL of 20picomolar DNA, yielding 7.2 × 10^9^ single-strand molecules per forward and reverse strands. The fed-to-bound molecule ratio on the flow cell is 722.7, meaning 1 in every 723 denatured molecules (0.14%) becomes bound and sequenced. The denatured DNA molecules, randomly bound in the flow cell, explain the poor intersection of haplotypes in the forward and reverse strands for singletons in highly flat-like quasispecies. Out of 723 denatured molecules, 722 were washed out of the flow cell. If we consider the ratio of single-strand molecules in the 1 ml denatured 20 pM library to flow cell-bound molecules, this value increases to 1204.4. This explains why the strand intersection strategy is not effective with flat samples unless denaturation occurs in situ.

### 3.2. Singletons

The distinctive trait of near-flat quasispecies is a high fraction of reads belonging to singletons, corresponding to haplotypes supported by a single read (Figure 1, Table 2 and Appendix A) and rare haplotypes, i.e., haplotypes with frequencies below 0.1% and with master haplotypes (the most represented sequence) at very low frequencies. This is illustrated in the form of a quasispecies fitness fraction (QFF) (Figure 2, Table 3 and Appendix A). In contrast, the regular quasispecies are characterized by the presence of a dominant haplotype at high frequencies, with low fractions of singletons and rare haplotypes.

The artifacts caused by amplification or sequencing errors are indistinguishable from the genuine, non-artefactual singletons and rare haplotypes. Nevertheless, when the samples compared are processed in parallel and sequenced in the same run, with high-quality scores, the observed differences may be confidently associated with different quasispecies states.

### 3.3. Hill Numbers Profile and Hill Evenness Profile

The Hill numbers profile (Appendix A) is the best for illustrating flat-like quasispecies due to its definition. A completely even quasispecies appears as a horizontal line, where all the haplotypes exist at exactly the same frequency, resulting in identical Hill numbers regardless of the order *q*. The Hill numbers profile is determined by two key indices: the number of haplotypes at *q* = 0 and the inverse of the master frequency at *q* = infinity. However, a perfectly even quasispecies is an idealization; there will always be a single haplotype with a higher frequency, termed “master haplotype”, despite its low dominance compared to the other haplotypes in the quasispecies.

The diversities represented by the Hill numbers of the order *q* can be transformed into evenness values by dividing them by the corresponding *D*(*P*,0) value, which is the maximum attainable value for any *D*(*P*,*q*). This transformation creates the evenness profile shown in Appendix A. Another alternative is the relative logarithmic evenness (RLE) profile [20], where the values are restricted to the 0–1 interval. The RLE values are calculated as the ratios of the logarithm of Hill numbers to the logarithm of the number of haplotypes, which represents the logarithm of Hill numbers at *q* = 0.

### 3.4. Indices of Evenness and Flatness

Indices of evenness (i.e., Shannon–Wiener and Simpson evenness) also serve as indicators of flatness, with the advantage that they are bound between 0 and 1. In this study, we used the ratios of Hill numbers of different orders *q*, with respect to the Hill number at *q* = 0. Importantly, when employing the Hill number profile to visualize the structure of quasispecies or the relationship between different quasispecies, the proposed indices are represented graphically, making their interpretation more intuitive and straightforward compared to the other indices. Appendix A shows the Hill index of evenness at *q* = 1, and Appendix A shows the index at *q* = infinity, where the ratio is computed on log10 values (Appendix A). Appendix A summarize these values for each quasispecies class.

The absence of significantly dominant haplotypes, an important property of flat-like quasispecies, is illustrated by the representation of the frequencies of the top 25 haplotypes (Appendix A). The corresponding quantification is provided by the ratio of the aggregated frequency of the top 25 haplotypes to the master frequency (Equation (4)) (Appendix A, Appendix A).

### 3.5. Genetic Space

The third condition of flat-like quasispecies, which involves occupying a significant fraction of the genetic space that is functional to the virus, may be evaluated using another profile; the fraction of reads corresponds to an increasing number of substitutions relative to the master haplotype (Figure 3).

In regular quasispecies, we observe significant fractions up to two or three substitutions, with a prominent fraction of reads for the master (m00). In flat-like quasispecies, we observe significant fractions at higher substitution multiplicities, with smaller fractions at lower multiplicities, indicating a much richer genetic landscape. The average number of substitutions per read relative to the master haplotype (Appendix A) provides a measure of this genetic plasticity.

### 3.6. Diversities, Evenness, and Sample Size

The implications of sample size on the observed values of diversity and evenness are discussed in the Appendix A.

## 4. Discussion

The failure of antiviral treatment has been typically associated with the selection of resistance mutations [24]. The high complexity of HCV quasispecies found in natural isolates [1,25] allows for the rapid exploration of the sequence space, leading to resistance-associated substitutions (RASs). The viral load, genetic barriers to drug resistance, and the fitness (replication capacity) of variants influence the selection of RASs during treatment, contributing to treatment failure [25,26,27,28,29]. For instance, viruses containing the Sofosbuvir (SOF)-resistant mutation at position 282 (S282T) of the NS5B gene exhibit very low fitness [30,31,32], and despite their selection during therapy, after stopping treatment in case of relapse, the viral mutants carrying this substitution do not predominate in the quasispecies. Other amino acid substitutions in the NS3, and especially in NS5A (e.g., Y93H in NS5A), dominate in some relapsed cases [4,33]. Surprisingly, in many cases and despite complete treatment adherence, after performing deep sequencing analysis, resistance substitutions are not detected at significant levels (>1%) in any of the DAA-targeted regions (own unpublished data from diagnosis of patients with HCV who were not successfully treated).

Some alternative causes of treatment failure include mutations at sensitive positions that have not yet been reported, residues affecting the protein structure, and mutations in positions causing allosteric effects altering the drug binding efficiency.

A different scenario arises when treatment failure occurs after the use of mutagens such as ribavirin in monotherapy. It is well known that viral persistence depends on the balance between replication fidelity and genomic flexibility, which facilitates the adaptation to changing environmental conditions [34]. An excess of mutations can drive a viral population to extinction [35], while too few mutations can cause extinction by limiting the population’s capability to adapt to a changing environment, such as escaping an immune response, replicating in different tissues, or transmitting from individual to individual.

Flat-like quasispecies may emerge in the presence of mutagens for two extreme and opposing reasons, resulting from the addition of selective pressure imposed by antiviral treatment (mutagenic treatment). The first is lethal mutagenesis, wherein any prominent haplotype is constantly pounded by a mutagen, generating less-fit haplotypes and contributing to a decrease in the production and frequency of well-fitted haplotypes. The target of such treatment is lethal mutagenesis, by which the virus increases a replication error above the error threshold, leading to extinction [36,37,38]. The second is a quasispecies with unusual genetic diversity that expresses a limited number of phenotypes and is highly resilient to environmental changes and treatments, with high average fitness [14,15]. The main factor distinguishing both these cases is the viral load. In the close-to-extinction case, it will be near zero, whereas in the resilient case, it will range from medium to high. The samples of the two HEV patients shown in this study, and taken as flat-like models, were produced in sequential treatments with a mutagen (ribavirin) discontinued before reaching a close-to-extinction situation with RNA negativization. Although lethal mutagenesis is plausibly described in quasispecies theory terms, there are external factors that may prevent it from reaching this final situation. The early discontinuations of treatment due to secondary effects of the mutagen on the patient, which result in relapses at appreciable viral loads, will cause the selection of new fitted variants produced during treatment, contributing to flatter quasispecies with an expected higher average fitness. On the other hand, the use of mutagens in the absence of efficient inhibitors could be insufficient to decrease replication to safe levels to limit the risk of the selection of sufficiently fitted variants, which could increase the quasispecies’ average fitness. The use of mutagens concomitant with medium to high viral loads leads to the selection of highly functional variants that are able to compete with the master phenotype, resulting in resilient quasispecies with expected high average fitness, as observed in [14,15].

Moreover, in addition to providing empirical evidence of in-host viral infections that are consistent with flat-like quasispecies characteristics, we propose methods to analyze and characterize these quasispecies, showing the limits of a comprehensive sampling. We also suggest that due to the distinct nature of this type of quasispecies, it must be studied using different approaches compared to those used on regular quasispecies. For instance, in studying DAA-treatment failure, the primary objective is the analysis and identification of mutations that are resistant to treatment [24,31]. Conversely, in highly diverse quasispecies exhibiting flat-like properties, fitness is paramount for resistance, and the main goal is to quantify the level of flatness and the portion of functional genetic space explored and occupied by the quasispecies. While the identification of RAS requires a minimal abundance filter and the intersection of both strands to mitigate false positives, the study of flat-like quasispecies demands a deeper view, where the absence of clearly dominant haplotypes and the presence of rare and very rare haplotypes collectively play a crucial role. Despite the evidence that coverage in the order of ~500,000 reads per amplicon might be insufficient to obtain a comprehensive representation of flat-like quasispecies (as discussed in the Results), the sequencing of two samples, S18 and S19, at a lower coverage, ~120,000 reads (which are replicates of S16 and S17, respectively) empirically demonstrated that a subsample of a flat-like quasispecies indeed exhibits flat-like quasispecies characteristics. In fact, at lower coverages, these samples display slightly higher values for the flatness indices.

Studying flat-like quasispecies while imposing abundance limitations is a daunting task, leading to the loss of a significant volume of information (such as singletons and other rare sequences). Consequently, we must be content with unavoidable technical and instrumental noise [15]. Since no cut-off in the observed, haplotype abundance was applied (due to the relevance of singletons), and we avoided the intersection of haplotypes observed in both strands to mitigate information loss, qualifying a viral quasispecies as flat-like relies on balanced experimental designs to ensure consistent levels of technical noise across all the compared samples. To achieve this, we recommend using samples of regular quasispecies as controls in each sequencing run when evaluating flat-like quasispecies. Additionally, we suggest including these controls in the wet laboratory steps and processing them in parallel with the new samples under evaluation to minimize the potential differential biases.

As shown in the Results section, distinctive traits in quasispecies with a flat profile were identified through the following indicators:A high proportion of reads corresponding to singletons.The limited overlap of haplotypes between the forward and reverse strands.A low frequency of the master haplotype, with a corresponding high value for its inverse, denoted as *D*(*P*,*∞*).An elevated load of rare haplotypes, particularly with a significant fraction of haplotypes below 0.1%.High values for the indices of evenness and flatness.A complex genetic landscape, with high values for the average number of substitutions per read compared with the master haplotype or, alternatively, relative to the consensus sequence.An elevated level of synonymity, reflected in a high ratio of master phenotype to master haplotype frequency.

The determination of what should be considered low, mid, or high values for these indicators remains unknown. However, the provided comparisons offer a guideline for classifying quasispecies into three levels: flat-like, near-flat, and regular. Further studies are necessary to obtain additional data, and a valuable source of samples may be derived from patients who have experienced treatment failure with mutagenic agents, as well as from patients with HCV who have failed treatments with direct-acting agents without evidence of resistant variants [1,24].

This study is clinically relevant because of these two reasons: (a) it predicts whether a chronic patient has developed a mature quasispecies that is highly resilient to treatments before undergoing any treatment, and (b) it gives information on the prevention of the formation of flat-like quasispecies with high average fitness due to treatment based on the analysis of sequential samples during patient follow-up. In such cases, caution should be exercised, and alternative clinical strategies should be considered whenever possible.

## 5. Conclusions

Using deep sequencing techniques, our study yields compelling evidence that the viral populations characterized by a very high number of different haplotypes, each presenting notably low frequencies and lacking a significantly dominant haplotype (referred to as a flat-like quasispecies structure), may contribute to antiviral treatment failure among patients who are chronically infected with HEV or HCV, particularly in cases without resistant-associated substitutions. Currently, we lack any clinical approach to address flat quasispecies. Here, we propose methods to identify them as a preliminary step towards investigating alternative treatment protocols.

## Figures and Tables

**Figure 1 microorganisms-12-01011-f001:**
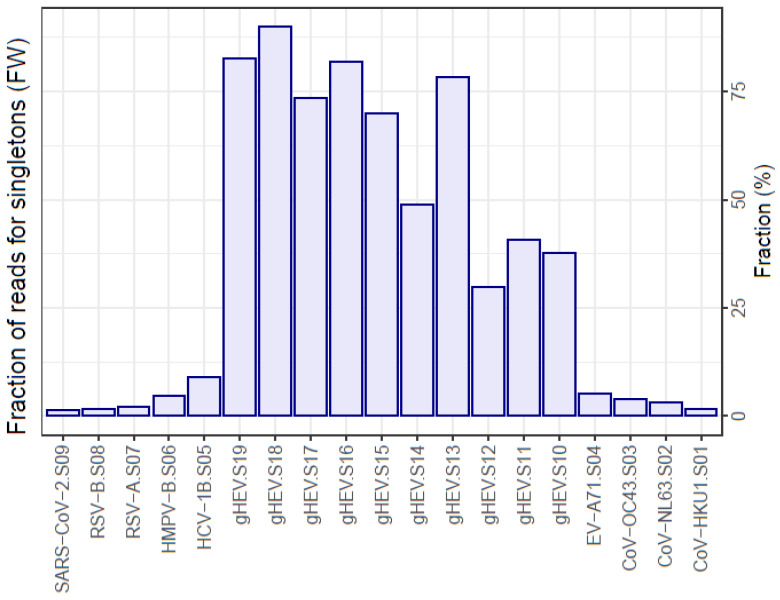
Fraction of reads in the quasispecies, corresponding to haplotypes supported by a single read. Flat-like versus regular quasispecies.

**Figure 2 microorganisms-12-01011-f002:**
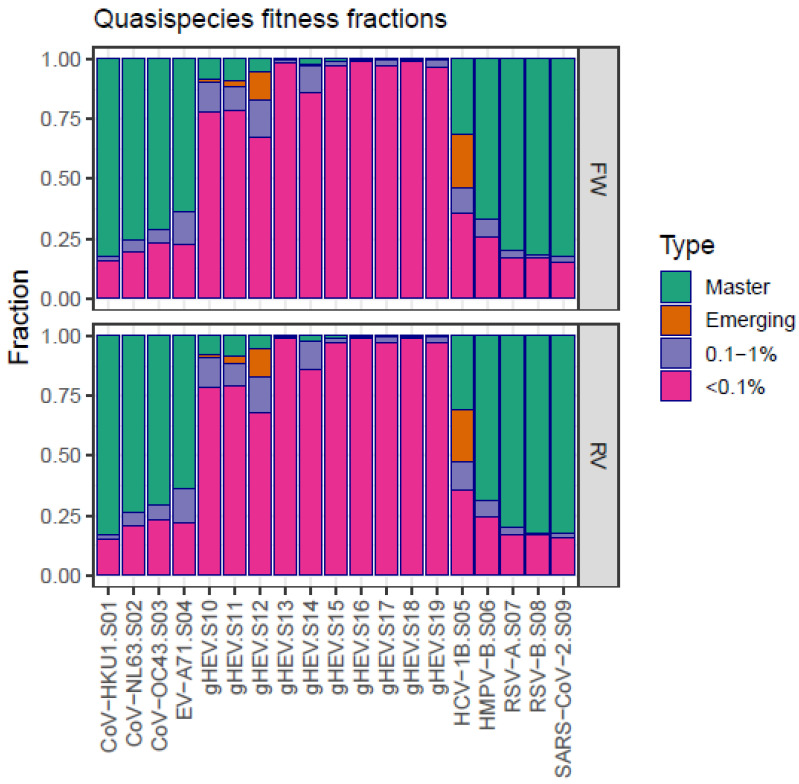
Quasispecies fitness fractions of flat-like vs. regular quasispecies.

**Figure 3 microorganisms-12-01011-f003:**
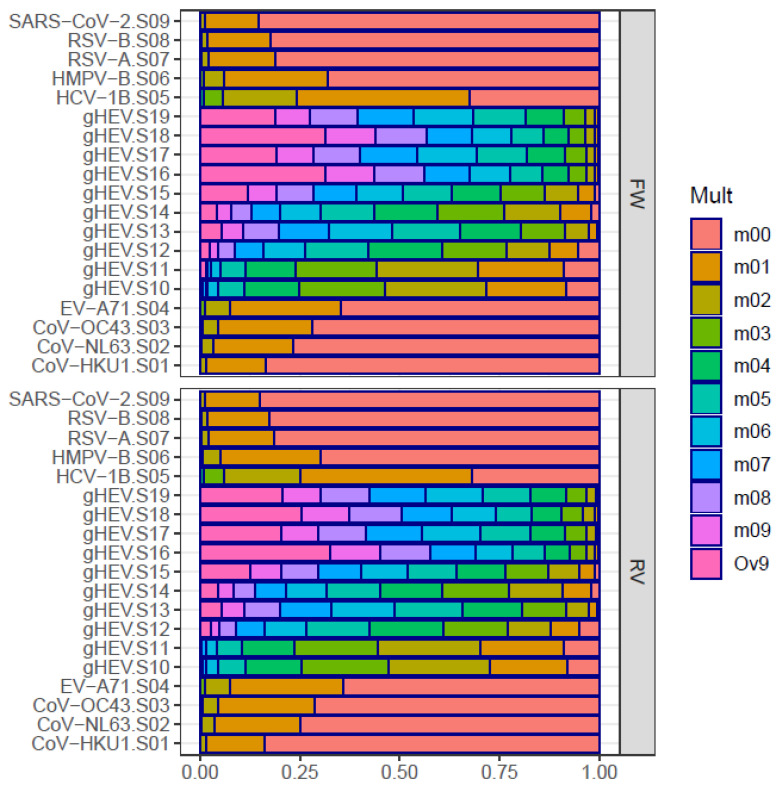
Genetic space. Fraction of reads corresponding to haplotypes at an increasing number of substitutions relative to the master haplotype in each sample: m00—fraction for the master; m0x—fraction of reads with x substitutions; Ov9—fraction of reads with over 9 substitutions relative to the master.

**Table 1 microorganisms-12-01011-t001:** Sample ID description. Samples are uniquely identified by the symbols ranging from S01 to S19. Samples S01–S04 and S06–S09 are from 8 different patients; sample S05 is from one patient with HCV chronic infection; samples S10–S13 and S14–S17 were sequentially taken from two different patients; S18 is a replica of S16, and S19 is a replica of S17 sequenced at a lower depth. For ease of interpretation, in some cases, the combination of a prefix indicating the virus and the unique symbols used as a suffix are displayed.

Postfix	Prefix	ID	Virus	Genomic Region	Primer 5′ End	Primer 3′ End	Reference Genome Genbank
S01	CoV-HKU1	CoV-HKU1.S01	Coronavirus HKU1	Spike	23,495	23,900	MH940245.1
S02	CoV-NL63	CoV-NL63.S02	Coronavirus NL63	Spike	21,794	22,220	ON554022.1
S03	CoV-OC43	CoV-OC43.S03	Coronavirus OC43	Spike	24,973	25,359	PP187318.1
S04	EV-A71	EV-A71.S04	Enterovirus A71	VP1	2844	3213	OR800939.1
S05	HCV-1B	HCV-1B.S05	Hepatitis C virus	Envelope	1985	2441	AF054249
S06	HMPV-B	HMPV-B.S06	Metapneumovirus B	Glycoprotein	6206	6634	MZ504966.1
S07	RSV-A	RSV-A.S07	Respiratory syncytial virus A	Glycoprotein	5924	6306	PP151405.1
S08	RSV-B	RSV-B.S08	Respiratory syncytial virus B	Glycoprotein	6415	6807	PP135061.1
S09	SARS-CoV-2	SARS-CoV-2.S09	SARS-CoV-2	Spike	23,347	23,713	MN908947.3
S10–S19	gHEV	gHEV.S10	Hepatitis E virus	Core	6373	6784	LC770331.1 and AB189070
		gHEV.S11					
		gHEV.S12					
		gHEV.S13					
		gHEV.S14					
		gHEV.S15					
		gHEV.S16					
		gHEV.S17					
		gHEV.S18					
		gHEV.S19					

**Table 2 microorganisms-12-01011-t002:** Summary of singleton fraction values observed in flat-like versus regular quasispecies.

	Flat	Regular
**Min.**	29.9	1.49
**1st Qu.**	42.0	1.66
**Median**	73.0	3.40
**Mean**	64.4	3.70
**3rd Qu.**	82.0	4.52
**Max.**	91.0	9.35

**Table 3 microorganisms-12-01011-t003:** Summary values of quasispecies fitness fraction values for flat-like (A) versus regular quasispecies (B). qff top: master haplotype frequency; qff_emer: fraction of reads for haplotypes above 1% and below master; qff_1_0.1: fraction of reads for haplotypes above 0.1% and below 1%; qff_0.1: fraction of reads for haplotypes below 0.1%.

**(A) Flat-like quasispecies**				
	**qff_top**	**qff_emer**	**qff_1_0.1**	**qff_0.1**
**Min.**	0.190	0.00	0.620	67.6
**1st Qu.**	0.281	0.00	0.872	79.0
**Median**	0.757	0.00	2.560	97.1
**Mean**	2.650	1.67	5.750	89.9
**3rd Qu.**	5.130	1.32	11.200	98.7
**Max.**	8.820	12.10	15.100	99.2
**(B) Regular quasispecies**				
	**qff_top**	**qff_emer**	**qff_1_0.1**	**qff_0.1**
**Min.**	30.7	0.00	0.927	15.4
**1st Qu.**	67.4	0.00	2.140	16.9
**Median**	74.8	0.00	5.040	20.2
**Mean**	70.6	2.46	5.660	21.3
**3rd Qu.**	82.2	0.00	7.260	23.3
**Max.**	82.8	22.40	14.200	35.8

## Data Availability

The genomic nucleotide sequences included in this study have been deposited into the GenBank repository database under the Bioproject IDs PRJNA1038697 (HEV), PRJNA662314 (HCV), and PRJNA1081281 (Respiratory viruses).

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
