# Peer review of "In-Host Flat-like Quasispecies: Characterization Methods and Clinical Implications"

_microorganisms, 2024, doi:10.3390/microorganisms12051011_

Round 1

Reviewer 1 Report

Comments and Suggestions for Authors

In this study the authors used NSG to compare In-host flat-like quasispecies among different viruses during treatment failure. THe authors used HCV, HEV, and other acute respiratory viruses (betacoronaviruses, enterovirus, respiratory syncytial viruses, and  metapneumovirus). THe authors reported that in case of HCV and HEV infection, the in flat quaspecies is characterized by numerous low-frequency-haplotypes with no dominant one.

The manuscript is not clear and rationale difficult to follow.

a) How authors compare chronic vs acute infections?

b) Did the samples from HCV and HEV are chronically or acute infection? Are these samples from coinfected patients?

c) The use of ribavirin is associated with resistance in both HCV and HEV, but it depends on doses, circulating gentoype, and immune status of patients.

d) THe authors used one strain from each virus, the conclusion can not be broad and applied to the whole virus family.

Comments on the Quality of English Language

Moderate language editing

Author Response

Reviewer 1

In this study the authors used NSG to compare In-host flat-like quasispecies among different viruses during treatment failure. THe authors used HCV, HEV, and other acute respiratory viruses (betacoronaviruses, enterovirus, respiratory syncytial viruses, and  metapneumovirus). The authors reported that in case of HCV and HEV infection, the in flat quaspecies is characterized by numerous low-frequency-haplotypes with no dominant one.

The manuscript is not clear and rationale difficult to follow.

  1. a) How authors compare chronic vs acute infections?

Answer:  Thank you for the comment. We understand reviewer 1 questioning the comparison of quasispecies from chronic samples and acute ones. The aim of this study has been to characterize different viruses in base of the “complexity” flatness of the viral population and try to find explanation to the fact that in some chronic infections the viruses are able to persist in the absence of resistance-associated substitutions. In the last 15 years, we have been characterizing viral populations from HCV treated patients, and we have been able to study HEV chronically infected patients that are not commonly found in the clinical centers, since HEV causes an acute self-resolving infection in immunocompetent patients. In our study, we have found flat-like and near-flat quasispecies in the HCV and HEV chronic patients, but not in regular quasispecies. Consistently, regular quasispecies were found in acute infections, that we were able to study form samples collected from patients with respiratory virus infections. For this reason, we decided to include and compare the QS from such diverse viruses.   See lines 99-105 in section 2 Materials and Methods of the clean corrected version: The samples were collected from eleven patients, one chronically infected with sub-type 1b of HCV [1], two chronically infected patient infected with HEV [15], and eight with various respiratory viruses. For the two patients with HEV, sequential samples from S10 to S13 were taken from one patient, and the samples from S14 to S17 were taken from an-other. No coinfections were observed in any of the patients included in this study. Addi-tional samples from the patients with chronic HCV infection were analyzed in the Sup-plementary Material (supplementary near-flat quasispecies examples).

  1. b) Did the samples from HCV and HEV are chronically or acute infection? Are these samples from coinfected patients?

Answer:  We thanks reviewer you this comment, since this point should be clarified. We have included a paragraph to clarify this point. See lines 99-105 in the clean corrected version (text included in comment a) “The samples were….examples”);  and legend of Table 1 (lines 115-119 in the clean corrected version): Sample ID description. Samples are uniquely identified by the symbols ranging from S01 to S19. Samples S01-S04 and S06-S09 are from 8 different patients; sample S05 is from one patient with HCV chronic infection; samples S10-S13 and S14-S17 were sequentially taken from two different patients; S18 is a replica of S16, and S19 ia a replica of S17 sequenced at a lower depth. For ease of interpretation, in some cases, the combination of a prefix indicating the virus and the unique symbols used as a suffix are displayed.”

  1. c) The use of ribavirin is associated with resistance in both HCV and HEV, but it depends on doses, circulating genotype, and immune status of patients.

Answer:  Reviewer is right. However, in this study, only HEV patient received ribavirin in three sequential time-points and using an increase dosage (citation 15 Gregori J et al, 2022, lines 621 to 623 of the clean corrected version). We agree with reviewer that it should be interesting to study the ribavirin effect using different dosage, comparing with different genotypes and the immune status of the patients, but this is out of our capabilities and probably it will require to use of an animal model. In our study of the chronically infected HEV patient, we think that the sequential study has the advantage that the control was the same patient, thus there are not differences in the immune status of the patients and no changes in the genotype. We hope that reviewer will agree with our point and accept the explanation.  

  1. d) The authors used one strain from each virus, the conclusion can not be broad and applied to the whole virus family.

Answer:  We agree with reviewer that more subtypes of viruses could have been included in the analysis, but we thought that comparing quasispecies with different population structure using a genomic fragment codifying for a surface protein subjected to similar immune selective pressures such as the spike protein of betaCoV, envelope of HCV, surface glycoprotein of other respiratory viruses and core protein of HEV (lines 99-105 in the clean corrected version ((text included in comment a) “The samples were….examples”);   and legend of Table 1 (text included in comment b) “Sample ID description... as sufix are displayed.”)  will be the best representants for other components of the viral families and enough for a breaking paper.

Comments on the Quality of English Language

Moderate language editing

Answer:  Thank you for the comment. English has been corrected using Author Services Language Editing solution provided by the Editorial. We have uploaded the “English-Editing-Certificate-80020”.

Reviewer 2 Report

Comments and Suggestions for Authors

The article is interesting and well-crafted, although it's hard to follow. To improve it, I suggest the following:

1. Lines 170-184, 195-211, 216-220, 225-227 need to be rewritten because a high percentage of similarity is detected.

2. Table 1, (title and legend) is edited differently from the rest of the text.

Author Response

The article is interesting and well-crafted, although it's hard to follow. To improve it, I suggest the following:

  1. Lines 170-184, 195-211, 216-220, 225-227 need to be rewritten because a high percentage of similarity is detected.

Answer:  Thank you for the constructive comment. The complete sections 2.6; 2.7 and 2.8 have been rewritten to eliminate similarities: See lines 184 to 212 in the clean corrected version. Moreover, in the previous submitted manuscript, section 2.5 was repeated in results. In the revised and clean corrected versions duplications have been eliminated. See lines 174 to 182: Material and Methods; 2.5 Deep sequencing “To acquire a comprehensive picture of quasispecies composition, our previous results have emphasized the necessity of conducting deep-sequencing studies with a very high coverage (over 1x105 reads per amplicon) and avoiding unnecessary abundance-based filters [17]. This is particularly crucial with mutagenic treatments, where their effect in the short term is primarily observed at the lowest haplotype frequency [15,17]. Nevertheless, in cases of extreme complexity, the simplification introduced by filtering all the haplotypes below a modest number of reads, i.e., from 2 to 5, could be acceptable provided that the resulting loss of information is limited..”; and lines 301 to 303 in the clean corrected version: ”Results 3.1 Sequencing coverage. “Deep sequencing was performed using an MiSeq Illumina instrument obtaining deep coverages ranging from 60000 to more than 200000 reads per strand, amplicon, and sample.”

  1. Table 1, (title and legend) is edited differently from the rest of the text.

Answer:  Corrected, thank you.

Reviewer 3 Report

Comments and Suggestions for Authors

Although this study is interesting and provides useful information, I have several concerns about the manuscript.

1. Please add more description regarding study methods in the abstract section.

2. The whole paper is difficult to read. Extensive English editing is needed.

3. Pleae focus on your own findings in the disccusion, avoding too many noises.

Comments on the Quality of English Language

The whole paper is difficult to read. Extensive English editing is needed.

Author Response

Although this study is interesting and provides useful information, I have several concerns about the manuscript.

  1. Please add more description regarding study methods in the abstract section.

Answer:  Thank you for the comment. We agree with reviewer that abstract is to cryptic. To solve this limitation, we have inserted a paragraph explaining the methods performed. See lines 29 to 36 in the clean corrected version): Quasispecies structured in a flat-like manner (referred as flat-like) are considered to possess high average fitness, occupy a significant fraction of the functional genetic space of the virus, and exhibit a high capacity to evade specific or mutagenic treatments. In this paper, we studied HEV and HCV samples using high-depth next generation sequencing (NGS), with indices scoring the different properties describing flat-like quasispecies. The significance of these indices was demonstrated by comparing the values obtained from these samples with those from acute infections caused by respiratory viruses (betacoronaviruses, enterovirus, respiratory syncytial viruses, and metapneumovirus).

  1. The whole paper is difficult to read. Extensive English editing is needed.

Answer:  Thank you for the comment. English has been corrected using Author Services Language Editing solution provided by the Editorial. We have uploaded the “English-Editing-Certificate-80020”.

  1. Please focus on your own findings in the discusion, avoding too many noises.

Answer:  We agree with reviewer that we discussion is too large and might be considered little bit further from the results presented. Our modest goal has been to enrich the manuscript and broaden the interest to a wide range of readers by redirecting the discussion to provide clinical applicability, such as giving explanations to treatment failures during antiviral treatment in the absence of resistance-associated substitutions or escape to mutagenic agents, despite the risk that it may appear as background noise. Anyway, following reviewer comment we have revised discussion and improved the narrative. For instance, first paragraph has been significantly reduced (see lines 432 to 447 in the clean corrected version): “The failure of antiviral treatment has been typically associated with the selection of resistance mutations [24,25]. The high complexity of HCV quasispecies found in natural isolates [1,26] allows for the rapid exploration of the sequence space, leading to resistance-associated substitutions (RASs). The viral load, genetic barriers to drug resistance, and the fitness (replication capacity) of variants influence the selection of RASs during treatment, contributing to treatment failure [26–30]. For instance, viruses containing the Sofosbuvir (SOF)-resistant mutation at position 282 (S282T) of the NS5B gene exhibit very low fitness [31–33], and despite their selection during therapy, after stopping treatment in case of relapse, the viral mutants carrying this substitution do not predominate in the quasispecies. Other amino acid substitutions in the NS3, and specially in NS5A (e.g.Y93H in NS5A), dominate in some relapsed cases [4,34]. Surprisingly, in many cases and despite complete treatment adherence, after performing deep sequencing analysis, resistance substitutions are not detected at significant levels (>1%) in any of the DAA-targeted regions (own unpublished data from diagnosis of patients with HCV who were not successfully treated).”

Comments on the Quality of English Language

The whole paper is difficult to read. Extensive English editing is needed.

Answer:  Thank you for the comment. English has been corrected using Author Services Language Editing solution provided by the Editorial. We have uploaded the “English-Editing-Certificate-80020”.

Reviewer 4 Report

Comments and Suggestions for Authors

The manuscript by J. Gregori et al. entitled "In-host flat-like quasispecies, methods and clinical implications" is devoted to the study of viral quasispecies from patients. The manuscript is well written. There are a few minor issues which are listed below:

Use square brackets for references.

Lines 60 and 63: Add references for these two sentences.

Add the aim of the sudy, it is not clear.

Line 87: Add what is DAA?

Table 1: For samples S10-S19, add what to which numbers in GenBank.

Lines 133, 137: Shorten Massachusets to MA

Line 145: Add Santa Clara, CA, USA for Agilent.

Line 288: Omit "(see"

Format the references as required by the journal.

Comments on the Quality of English Language

The English is good enough to be published.

Author Response

The manuscript by J. Gregori et al. entitled "In-host flat-like quasispecies, methods and clinical implications" is devoted to the study of viral quasispecies from patients. The manuscript is well written. There are a few minor issues which are listed below:

Use square brackets for references.

Answer:  Done. Thank you.

Lines 60 and 63: Add references for these two sentences.

Answer:  Done. See line 69 with the citations 12 and 13 in the clean corrected version.

Add the aim of the study, it is not clear.

Answer:  The objective of this paper was to devise tools for characterizing quasispecies structure, aiming to elucidate naturally occurring phenomena, such as the ability to evade antiviral treatments, without selection of specific antiviral resistance substitutions.

Line 87: Add what is DAA?

Answer:  DAA has been defined see lines 92-93 of the clean corrected version: “...direct-acting antiviral (DAA)-“

Table 1: For samples S10-S19, add what to which numbers in GenBank.

Answer:  Thank you for the observation. We aligned two GenBank sequences with samples S10 to S19, which are sequential samples from two HEV infected patients. Consequently, we believe that additional HEV GenBank references are unnecessary. We hope you will consider our viewpoint.

Lines 133, 137: Shorten Massachusets to MA

Answer:  Done. See lines 150 and 154 in the clean corrected version.

Line 145: Add Santa Clara, CA, USA for Agilent.

Answer:  Done. See line 162 in the clean corrected version.

Line 288: Omit "(see"

Answer:  Done. See line 282 in the clean corrected version.

Format the references as required by the journal.

Answer:  Done, thank you.

Round 2

Reviewer 1 Report

Comments and Suggestions for Authors

No further comments

Comments on the Quality of English Language

Language is fine

Reviewer 3 Report

Comments and Suggestions for Authors

The authors had addressed all my concerns.